# "They Finally See Me, They Trust Me, My Brother's Coming Home" Recognising the Motivations and Role of Siblings Who Become Kinship Carers

Lorna Stabler 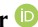

Children's Social Care Research and Development Centre, School of Social Sciences, Cardiff University, Cardiff CF24 4HQ, UK; stablerl@cardiff.ac.uk

**Abstract:** Despite a widespread focus on grandparents, a large proportion of kinship care in the UK is provided by older siblings. What drives older siblings to become kinship carers, and how this might differ from other kinship carers, is not well represented in academic literature. In this study, narrative interviews were carried out with thirteen adults across England, Scotland, and Wales who had experience being the main carer for their younger sibling(s) when their parents could not care for them sufficiently. The narrative method elicited holistic accounts of participants experiences of being a sibling carer, and the analysis generated three groups of narrative accounts highlighting how and why some sibling kinship care arrangements come about—with siblings wanting to bring their younger siblings back into their family; siblings trying to keep their younger siblings in their family; or older siblings stepping in to fill a gap in parenting at home. The paper draws on the narrative accounts of participants to build the groups, presenting an illustrative narrative account to represent each group. Importantly, these accounts demonstrate how becoming a kinship carer as an older sibling may, or may not, be recognised or fit into wider narratives of what becoming a kinship carer looks like. It is hoped that these accounts will prompt practitioners and policymakers to look more closely at the role of siblings when considering who is and who should be involved in deciding how to support children to remain within their family network.

**Keywords:** kinship care; sibling relationships; social work; narrative inquiry

## 1. Introduction

There has been an emphasis in recent years on sharing stories to gain recognition of kinship care, including highlighting the extent to which it is happening in the UK and the struggles that kinship families face. This push for recognition reflects how policy and practice not only rely on evidence but also on living examples to highlight what is needed and why. Estimates from the UK Census point to a significant but decreasing population of sibling-headed families [1–4], although the actual number of siblings providing care to children is unknown due to how population level statistics are collected [1]. In the 2021 census, for example, siblings aged 18 and over were only assumed to be potential carers when there was no grandparent or other relative in the household. The reality is that an older sibling might be caring for both a younger sibling and a grandparent, but this would not be identified in the census. The experiences of sibling-headed families have not been widely documented in the academic literature, with an emphasis on other care arrangements at different life stages and with different relationships to the child, particularly grandmothers [5].

The situations of sibling-headed kinship families may be very different from one another and from those of other kinship carers, due to their being, on average, younger than other kinship carers, often having shared parents with the children they are caring for, and being less likely to have experience as parents [6–9]. Moreover, little is known about the caring roles that siblings take on in their families, where there may be gaps in parenting.

Providing these carers with a platform to express their experiences could be an important mechanism for this group to be considered in practice and policy. Stories powerfully communicate real-world experiences in the context of social work [10] and influence policy [11].

This paper focuses on exploring the experiences and motivations that lead siblings to become kinship carers and some of the challenges different routes to becoming a kinship family can present.

Few studies have focused on sibling kinship carers in the UK. One qualitative study was found that focused solely on sibling carers (*n* = 12) [6]. In others, sibling kinship carers represent a subset of participants (i.e., [12] includes 6.25% (*n* = 5) siblings), while the majority of academic papers that include kinship carers do not disaggregate by relationship (i.e., [13]). Research from elsewhere in the world (i.e., [7–9]) resonates with findings in the UK, but these studies do not reflect local policy and practice within the UK.

How kinship families are formed—in response to which issues and with or without the intervention of children's services—can impact later support needs and whether or not these needs are met, due to eligibility criteria applied to accessing support. The kinship family may be assembled quickly in response to a crisis, with the new family needing to construct new roles and responsibilities [14]. A few studies identified reported that some kinship carers felt unprepared for the role, feeling coerced into agreeing to become carers to avoid the child going into care or being adopted [15–17]. Some kinship carers report not having fully understood what becoming a kinship carer would entail before taking on the role [16], assuming they would be able to get support for the child in the future [18]. For sibling carers, their age may also impact their engagement with services and how they are treated by professionals. Some studies have noted that sibling carers felt they were not able to attend school meetings [12] or did not feel that children's services took them seriously, questioned their capability to parent, and did not give them support, advice, or information [6] because of their age.

The transition to kinship care might not always be as clear for siblings as in some other kinship families. Gaps in the parenting received by sibling groups can lead to a desire from older siblings to do a better job of parenting than they feel their parents have done [14]. Siblings in these situations may sometimes take on a caring role for a long time before officially becoming their full-time carer, sometimes taking on the role of carer throughout their own childhood [6]. Despite this, the transition to full-time care of a child is likely to be a big adjustment for sibling kinship carers, as they are less likely than other kinship carers to have parented a child, particularly an older child; they may be on average younger than parents in the population [2], and they may not have anticipated such responsibilities at that point in their lives [6].

In two unpublished studies of informal kinship care [12,19], around one-third of children in informal care had experienced parental bereavement. Where the sibling kinship carer has shared parentage with the child they are caring for, they may have their own experience of parental bereavement. While it is noted that kinship children who have experienced bereavement will likely need specialist support and that carers may need training to deliver this [20], there are no specific bereavement services for sibling kinship households that have experienced bereavement. Moreover, while research indicates that carers may also be suffering bereavement and loss leading to the kinship arrangement [12,21], there is little provision to address this through the current provision of kinship support.

The studies highlighted indicate that sibling carers, like other kinship carers, become carers in a multitude of ways, often in response to crises and gaps in parenting experienced by a child. However, for sibling kinship carers, the transition to being a kinship carer can happen alongside the carer's own experience of insufficient parenting or bereavement. At the same time, the age and life stage that sibling carers are at can mean they do not get the right advice or support to transition into the role. However, little is known about who sibling kinship carers are, how these families are formed, or what their experiences are of becoming kinship carers. This paper aims to explore in more depth these experiences and

encourage close reflection on the role siblings should have when considering supporting children to remain within their family network.

This paper presents one element of a wider doctoral study, focusing on the support needs of sibling-headed kinship families from the perspective of the older siblings. The inspiration and desire to study this topic came from my own experience as a kinship carer for my younger brother. When I read research about kinship care and saw awareness-raising campaigns by charities, I did not see my experience represented.

## 2. Materials and Methods

This study sampled and recruited individuals who defined themselves as carers for their sibling based on their self-identification as kinship carers, and their own definition of a sibling. Like all children, those in kinship care do not understand their sibling relationships to be bound by biology [22]. Sibling-headed kinship households are structured around different bonds, such as step-siblings or siblings with only one parent in common, rather than 'blood' which could impact more on the choices of grandparents when making decisions about which children they could care for. Rather than focusing on a specific legal order or imposing definitions on sibling kinship care (i.e., no parent in the household, only 'full' brothers and sisters), this study included anyone who had for a period of time defined themselves as having been the main carer for someone they had a sibling-like relationship with. This approach aimed to gather a broad sample of sibling-headed kinship families with a range of entry points into becoming kinship families. Carers were recruited across Scotland, England, and Wales as care arrangements cross these borders; however, policies around support differ across nations. Recruitment was carried out through established support groups and organisations working with kinship carers, social work teams, and also through social media.

This paper presents the analysis of narrative interviews carried out with sibling kinship carers (*n* = 13). Although originally 20 participants were sought, there was difficulty identifying more potential participants. Narrative interviews were chosen to enable the experiences of sibling-headed kinship families to be explored inductively rather than ascribing preselected theories. When seeking to include a population such as sibling kinship carers that is often hidden and marginalised, the opportunity to speak freely and focus on areas that the kinship carers felt were important was essential.

Different approaches, strategies, and methods can be used within the framework of narrative research [23]. The approach used was designed to enable participants to identify what they viewed as key events, players, and circumstances in their journey to becoming and being kinship carers. Accordingly, a naturalistic narrative stance was taken [24], asking people to tell the story of their experience. Using this approach can quieten the researcher's voice and enable participants to lead and direct the interview conversations, particularly where they have engaged in a pre-task to prepare.

The narrative interviews centred on the experience of raising a sibling. Participants were given an optional task to encourage them to reflect and prepare before the interview to help them consider what they may want to share. Using the method suggested by McAdams [25], the task asked participants to think about their life in five chapters or events, including before and since they became carers for their sibling, listing key events and 'characters' relevant to the period covered. The tasks were not shared with the interviewer but were used by the interviewee to guide their responses.

Interview guides were developed by the author in consultation with two senior academics (DM and RE) and with feedback from two pilot interviews. Participants were asked an open narrative question: "I would like you to tell me the story of your life before and since you became a carer for your sibling. Please start wherever you feel is relevant". This open question was designed to enable their story to be told in a way that felt appropriate for them [26]. Prompt questions were kept to a minimum throughout the main narrative and were used for clarification. The narrative method aims to make the experience natural, reduce the cognitive load on participants, and help keep a focus on

what I (in the research) am interested in—the lives of sibling-headed kinship families [24]. Interviews lasted between 53 min and 147 min, with most taking around 2 h.

The conduct of the research was given approval by Cardiff University School of Social Sciences Research Ethics Committee (ref: 4093). Considerations were given to the potential distress of the participant. Due to the topic of the research, it was important to acknowledge that participants may experience some emotional distress through talking about difficult experiences. This was one of the main reasons for choosing narrative interviews for sibling participants. Participants were given preparation time and an option over how and where to participate, with interviews conducted online or in person depending on the preference of the participant. Breaks were encouraged. Participants were provided with resources to access if needed and offered a debriefing after the interview. Interviews were audio recorded, transcribed verbatim, and anonymised.

A thematic narrative analysis was conducted, identifying common elements to theorise across cases, where the "...emphasis is on...the events and cognitions to which language refers (the content of speech)" [27] p. 58), with a focus on the temporal events within the accounts [28]. Transcripts were prepared manually by the author and read for familiarisation. Initial codes were generated while reading through the full transcripts a second time, assigning labels [in Microsoft Word documents] that captured the understanding of the meaning of a segment of the data rather than labelling with short or theory-led codes [29]. Second coding is noting key events and phrases that were used to develop timelines for each account [30] and to plot the events chronologically (called restoring in narrative analysis) [27]. These codes were compared across accounts to draw out similarities and differences in important stages and events in kinship families' lives, consolidating codes into themes across cases. Analysis was carried out by the author, with the selection of transcripts and coding reviewed by one of two senior academics (DM and RE) to ensure rigour, and a reflective diary was kept by the author throughout. The analysis was shared with participants where possible in different ways throughout the analysis process (including member-checking interviews and sharing of the write-up).

## 3. Results

### 3.1. Participants

Thirteen older siblings took part in interviews (Table 1), six of whom are currently identified as the main carers for their sibling, with seven having previously been in this role. Two had a legal permanency order for their siblings; three were registered foster carers; and eight had no legal order in place. The majority ($n$ = 12) were female and 1 male, with only 2 identifying as not white.

**Table 1.** Selected demographic characteristics of participants.

| Organising Theme | Country | Kinship Carer and Age When Kinship Began | Age of Younger Siblings at the Start of Kinship Care | Legal Order |
|---|---|---|---|---|
| Bringing the family back together | Wales | Joanne (25) | 8 and 9 | Special Guardianship Order |
| | England | Sally (25) | 13 | Care Order |
| | England | Stacey (21) | 15 | Care Order |
| | Wales/England | Emma (22) | 6 and 7 | Care Order |
| | Scotland | Kara (20) | 1 | Adoption Order |
| Keeping the family together | England | Anna (22) | 13 | No Order |
| | Wales | Marcie (27) | 11 | No Order |
| | Scotland | Izzy (22) | 14 | No Order |
| | England | Claire (22) | 14 | No Order |

| Organising Theme | Country | Kinship Carer and Age When Kinship Began | Age of Younger Siblings at the Start of Kinship Care | Legal Order |
|---|---|---|---|---|
| Keeping the family together | England | Jade (25) | 6 and 8 | No Order |
| | Scotland | Laura (18) | 14 | No Order |
| Stepping in to fill a gap | England | Hasan (15) | 5 | No Order |
| | England | Kelly * | Four siblings * | No Order |

\* ages unclear as caring roles fluctuated.

### 3.2. Groups of Narrative Accounts

The narrative accounts had common characteristics [31], which allowed them to be grouped together to present some typical journeys to becoming a sibling-headed kinship family for these participants. These are not all possible routes that could have been taken, but they give examples and a basis for considering what other journeys there may be. The accounts were driven by time, expressing a journey and the events leading up to the formation of the kinship family. The accounts were told in relation to a prompt; therefore, overemphasis and exclusions based on the interest or inferred interest of the researcher are to be expected [32], especially in this case where participants and the interviewer had a shared experience of being sibling kinship carers. These accounts do not tell a life story, just the events the narrator included in their telling.

Narrative accounts were categorised by their 'organising theme'[1] [33] (see Figure 1). The organising theme was the motivation of the protagonist (the sibling kinship carer participant)—why they became a kinship carer. The sub-characteristic that categorised the accounts was the protagonist's position in relation to their siblings—whether they were separated, where they were living, whether or not they were in care, and what role they played in each other's lives. This paper focuses on the motivations of the carers, although other themes are touched upon.

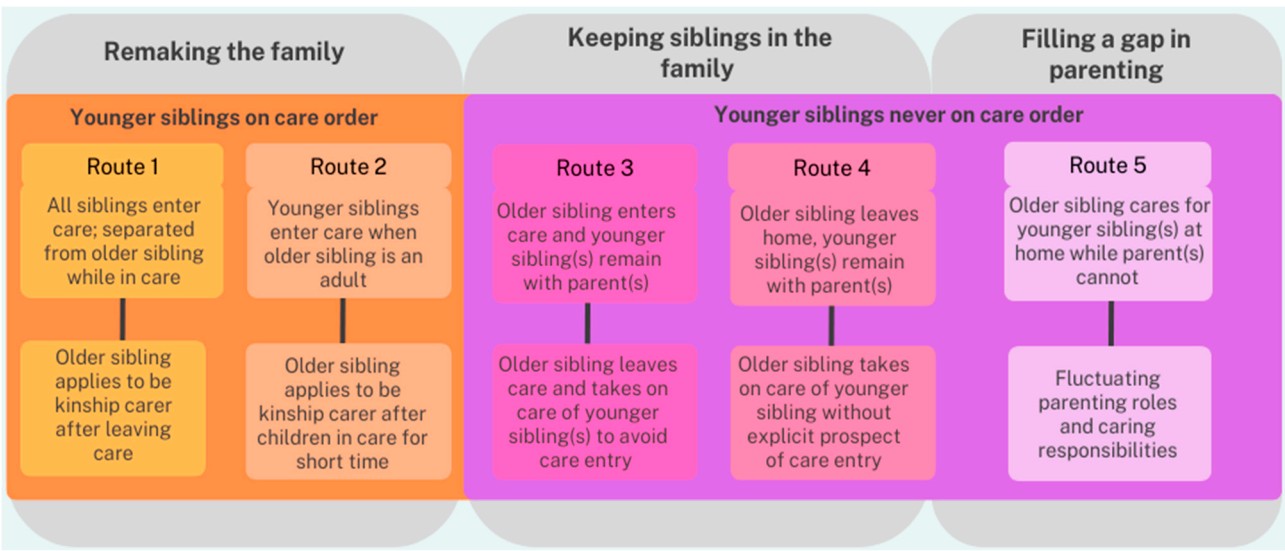

**Figure 1.** Categories of narrative accounts generated through narrative analysis.

There are many ways that narrative accounts could have been grouped. The choice made reflects and problematises the common categorisation of 'informal' and 'formal' kinship care arrangements, which does not capture the complexity of how kinship arrangements come about. Categorisation by the motivation of the protagonist centralises their agency, experiences, and wishes—centring their importance in each other's lives. This is



a choice I made in response to the fact that siblings are often missing from more widely shared narratives of kinship and foster care.

### 3.2.1. Group 1: Remaking the Family

Group 1 included five narrative accounts—Joanne [2], Sally, Stacey, Emma, and Kara. The organising theme based on these carers' motivations was remaking the family after separation, sometimes also avoiding further, more permanent separation of the siblings. This involved bringing the child siblings back into the care of their family after they had been placed outside of the care of their relatives. Across the narrative accounts, elements of separation and remaking of the family unit were evident. However, within this group, this involved active negotiation with formal systems to realise this end, including some form of assessment and legal order so that the older sibling could become the carer for their younger sibling.

There were two routes evident through which these kinship families were formed. On the first route (route 1 on Figure 1), whose narratives include Stacey and Sally, the younger siblings had been in care for several years while they themselves were in care, where they had been separated from their younger siblings. A desire to be back together throughout that time was communicated by older and younger siblings. Sally's younger brother had regularly been requesting in his review meetings to live with his older sister. Stacey's younger brother had been moved from his long-term foster home into a residential care home. For both of these sibling groups, the older sibling spoke with their personal advisor [3] and the younger siblings' social worker, and they were encouraged to apply to become foster carers.

The second route included Joanne, Kara, and Emma, who were adults when their younger siblings entered the care system. These siblings were informed that their siblings had entered care and were involved in discussions with authorities about the next steps, such as supporting a parent to be able to have the children back at home. Joanne and Kara put themselves forward to children's services as an alternative carer when it became clear that the child living with their parents was not going to be an option. For both families, the route pursued involved a legal, permanent route. Emma applied to be assessed as a foster carer [4] when she found out that her younger sisters had been placed in foster care in a different region and their mother had gone missing.

The illustrative narrative account tells of how Joanne came to realise she would be the one to take on the role of bringing the siblings back into their family.

> Joanne: *"I still don't know exactly how or where or why that happened but somebody had called [their nanny] and she called me. I called [then younger sibling's father] and said, you know 'what the hell's going on?' I went straight out there. I said [to social services], 'look, what needs to happen to get the children back?' They said, well, 'he [the younger siblings' father] needs a house and a job. And to stop drinking for three months.' I thought, like. . . (sigh) I'm pretty sure I've only ever seen him sober three times in the past 10 years. Like properly stone cold sober.*
>
> *I think I kind of knew, in my heart that it probably wasn't something that he could do, but I just hoped that he could, because I thought this has got to be his rock bottom, surely. I got him a house and a job. He had [three months] to stop drinking.*
>
> *I managed to speak to the children every two weeks. They only had each other. I was allowed to phone every two weeks. It's very difficult having a conversation on the phone with children. But obviously, I told them about nan [grandmother], and [their older brother]. So I think in their heads, they knew they had that family. . . somewhere.*
>
> *I think I thought maybe, the kids being in care, I know they're safe. I know they're fed. I know they are going to school. At least the daily needs were taken care of. I thought, maybe if I just buy some time, they'll be fine in there for six months, a year or something. I think that it was actually three months was a bit of a shock. I don't think I'd had time.*

*There was no preparation process. There was no nine months of pregnancy to get your head around the process.*

*I got the phone call that said they're going to be put out to adoption, if no one in the family can take care of them. In my head, I was like 'family means me'. I know that means me. Well, I'll do it then. It was quite bizarre how not shocked I was by what came out of my mouth I think. The two questions I asked were, whether I could still see them, and whether they'd be kept together. The answers to both of those questions were they couldn't guarantee either of them. For me, it was just completely unfathomable, I might not be able to see them, and they might not even have each other.*

*Before they [the youngest siblings] were born, it was me and [my brother] that just had each other. Then there was [the youngest siblings] who just had each other, and [the youngest] used to follow [the older one] around like a puppy bless her. I didn't doubt that was the right decision. I've never doubted that it was the right decision.*

*I don't really know what I thought was gonna happen then. I signed the papers, probably within 24 h—like a registration of interest or something. Then I thought, 'Well, I'm their sister. They know me, I've met them. I'll just get them'. I somehow thought it was just going to be like that.*

*But as it was, it wasn't at all like that. It took nearly two years".*

Joanne's account illustrates the theme of the importance of sibling relationships and how this impacted her decision-making. The importance of being together as siblings underpinned the story, with Joanne relating how valuable having a brother was to her in her childhood to the role that the younger siblings played for each other. This need for the siblings to have each other in their lives was a strong motivating force for the formation of the kinship family. The shared biological, cultural, and family histories common to the siblings were also highlighted as reasons why they should be together.

Sibling connections were also prominent in Sally's account, where she talked about wanting to get her siblings out of care as soon as she herself left.

Sally: *"I was like 'So now that I've got my own flat, why can't they just come live with me? They are my brother and sister. You don't own them. They're my family, that is my blood in there, not yours'".*

Recognition of a key person or people in seeing the motivation and ability of an older sibling was often a pivotal part of the narrative accounts in this group and potentially shaped the older siblings' own confidence in their ability to bring their siblings back into their family. This was featured in Joanne's longer account and also in Emma, Kara, and Sally's.

Sally: *"It was them [brother's foster carers] that suggested 'look why don't you get in touch with his sister. It is something she's always wanted, it is something he's always wanted. Why are we not pushing for this' I remember, not long after this picture [photo of carer and younger sibling] was taken, getting a letter. I literally ran out my front door. Anyone would have thought I'd won a million pound. I ran out the front door and I just did a full lap of the street, all the way around the full street. I stood in my garden doing star jumps. I don't understand why I did that. But it was this feeling of 'oh my god, they finally see me, they trust me. I'm responsible, that they're gonna give me my brother. My brother's coming home'".*

Emma described how sharing her story was difficult during the assessment, especially having to detail events from her early childhood. However, the assessment also provided an opportunity to show how motivated she was to bring her siblings into her care. She highlighted how she felt the recognition was her motivation for the outcome of the panel when deciding whether she could become a foster carer.

Emma: *"I think once we went through the whole process of what I'd been through, especially in the nitty gritty detail that you have to go to, she [assessing social worker]*

*was a bit more understanding of why. Right, I get why she wants them. I fully understand why she wants them. I understand that, despite her being so young, it just makes complete sense. She was really understanding and she was perfect. When we went to [the fostering] panel, there were discussions brought up about the abuse when I was a child. It was insinuated in a way that because of that, it might bring up memories for myself. It might cause me to be incapable. Which I was like…'urrr…mmm. Really like what? So you're going to essentially discriminate, because I've experienced that as a child.' But luckily, as [the assessing social worker] had said, there was a previous looked after child on panel. I don't know if she had experience with abuse or what but she basically said 'it's wrong for us to factor that in.' Every question they'd asked me, I'd answered. Apparently her impression of me was just that I wanted to get my family back together*".

The recognition in these accounts contrasts with the findings of Roth and colleagues [6], where sibling carers felt they were not recognised by services. However, these experiences of recognition stood out due to the contrast often with not being supported or taken seriously at other points. Motivation to bring the family back together was often contrasted with the timelines and processes within services that were not seen to facilitate strengthening family identity or connection. While assessment processes formed part of this, a much more systemic lack of support for maintaining sibling relationships was highlighted in this group of narrative accounts. For example, Emma talked about how, despite having been successfully approved as a kinship foster carer for her sisters, they remained with their foster carers for months, where they were taught a language that was not their family language. Stacey, Sally, and Emma talked about limited opportunities to spend time with their siblings in environments that allowed them to rekindle their family life, with 'family time' being limited and taking place in contact centres. Other narrative accounts highlighted how a lack of information and preparation limited opportunities to give siblings an understanding of their journeys and experiences and develop a shared story together.

There was also a lack of representation of sibling or younger carers in wider narratives about taking on the care of children, which could be a barrier to older siblings recognising themselves as potential carers.

Joanne: "*I hadn't considered the possibility because I just didn't see myself as somebody who was grown up enough to have children. It never crossed my mind to have children of my own. I also had this vision of the people who adopted children are people who only have to work part time and have lots of money, and they have big houses, with walls and driveways and things. I guess I didn't have that kind of visual reference tool that it could possibly be me*".

These accounts highlight not only how motivated siblings might be to step in to bring their younger siblings back into the care of their family but also the barriers that might exist to being identified as a potential kinship carer.

3.2.2. Group 2: Keeping the Siblings in the Family

Six narrative accounts developed this group—Anna, Marcie, Izzy, Claire, Laura, and Jade. In this group, there were two routes identified in the formation of the kinship family. For Claire and Laura (route 3 in Figure 1), a crisis situation led to the younger sibling no longer being able to remain in the care of the parents. Anna, Marcie, Izzy, and Jade (route 4 in Figure 1) actively intervened due to concerns about the safety of their siblings, with Anna, Marcie, and Izzy's siblings actively involved in wanting to leave their parent's care.

The circumstances preceding the kinship care arrangement included the death or serious illness of a parent, a lack of parenting capacity due to drug or alcohol use, domestic violence, and suspected sexual abuse. There was an implicit or explicit potential for the younger sibling to have to enter the care system due to a lack of appropriate family members to care for them.

The narratives in this group challenge the idea of 'formal' kinship families as those with social work involvement and 'informal' kinship families having made a kinship

arrangement themselves. While in two of these families (Anna and Izzy), there was no social work involvement at all, for the rest, social work intervention had in some way preceded or led to the care arrangement; it just did not lead to a legal order that placed the child with the older sibling. For example, for Jade, the local authority asked her to care for her siblings for an unspecified amount of time while a child welfare assessment was carried out, and for Laura and Jasper, they were told they would be eligible for support, only to find out later that it was not. For others, social work involvement had not directly preceded the kinship care arrangement, but there was often awareness by social services of the situation, like for Claire, who was still under a leaving care team [5] when she became a kinship carer for her brother.

In the illustrative narrative account below, Laura talks about how she had been caring for her brother in the years before a crisis led to her role being recognised as that of a kinship carer. Her role, however, not being subsequently formalised, led to a lack of support.

Laura: "*After my dad died, things started going on downward spiral in the family. My big [older] brother's behaviour, in the family home and my mum's addiction and mental health. My older brother left. So at that point, it was just me and my younger brother, and my mum. Mum was there but she wasn't there mentally. She wasn't well. She would be drunk and passed out on the sofa. I would make our tea and just make sure he was alright, just check in with him really.*

*It came to a to a stage where I'd gone away for the weekend to a camping festival. I must have been 18 at the time. My first camping festival with my friends and my mum and my brother were in the house on their own that weekend. When I came back, my brother was like 'mum's been really bad, thank god you're back, mum's just not right.' She was not in the best health and had been declining ever since my dad died, the alcohol, not eating. She wouldn't go out of the house on her own.*

*Because it was a decline over so many years, and we were so used to seeing her every day, I don't think we realised how unwell she was. We were like, 'well, she is like this all the time.' The thing that was different this time with her health was she was falling over a lot. But she wasn't drunk. She was very confused. She would say to me, 'how was your school?' I'd left school years before. But we weren't really sure. We were like, 'maybe she's just drunk, I don't know what's wrong with her.' She fell over one time and I seen her on the floor, rolling about and I was like, 'mum, what are you doing'" She was like, 'Oh, I'm picking something up.' I knew she fell over, but she just didn't want to admit it, so she was pretending to pick something up off the floor. I didn't know what to do.*

*Luckily, my friend's mum who was a nurse, she said, 'I'll just take you [to hospital] in the car.' I don't know how we managed to persuade my mum in the car but we managed it. Got her up to hospital. They sectioned her straight away and said, 'you need to stay in here, you can't leave.' They didn't know exactly what was wrong with her at this time. But later on, they diagnosed her with alcohol related brain damage.*

*I think it was the hospital that would have made a referral to social work. It was children and families team and a social worker came around to the house. It wasn't really a good experience. It felt very informal and chilled. She came to do an assessment to see if I was a fit carer to look after Jasper and what we wanted to happen. We had a wee chat and it was like, 'oh, well, what do you want to happen? Is there anyone that can look after [youngest brother]? Or can you just keep looking after him?' They were asking, is there anyone else? I was like, 'no, like, there's no anyone else.' The social worker just said, 'oh well if you don't want to look after him, there's no one else in the family that can look after him. So the other options would be to go to a children's home.' I was like, 'whoa, like, no, no, like, I'll just look after him here. This is as much his house as it is my house, this is the family home, why would he go somewhere else?'*"

As can be seen in Laura's account, within this group, 'becoming' a kinship carer was often an extension of the caring role the sibling was already playing in their younger

sibling's life. The account also shows how the decision to become a kinship carer occurred in a time of crisis for the family, where older siblings are balancing caring responsibilities. This was echoed by Claire, where her own experience of the care system played into her decision to step in to avoid her brother having a similar experience:

> Claire: *"My brother's dad had got very sick very quickly and I had made a promise on his deathbed that I would look after his son. Which I held myself to for a very long time. In my eyes, I did not want him to go into the care system. I just knew he wouldn't survive. He had enough issues anyway. I was like, 'No way is he having that instability'"*.

As indicated in other research, how decisions are made has implications for the support to which children and carers are entitled [17]. Stepping in to avoid the child entering care could lead to no eligibility for support for these families, despite the circumstances and needs of the child not necessarily differing from children who are on a care order. Siblings talked about the struggles that they experienced trying to get the basics needed for childcare in the early days, such as school uniforms and spare beds. But they also talked about how the lack of financial support made them feel unrecognised and unvalued.

> Laura: *"If there was some form of financial support, and that would help with arguments, bills, and also it would have made me feel a bit more valued. The role I'd taken on, I mean, I really would rather not be arguing with my brother all the time. But if at least, it wouldn't even be about the money. Even if it was like, a small amount, it's the recognition of what you're actually doing...even like a council tax reduction, or like, you know, like, there's other things you can, like, yeah, like a carers card money off, like trips out and stuff"*.

In some cases, there was no involvement with children's services at all, but the older sibling came to the understanding that the child could not remain with the parent(s) and there was no alternative option. In these cases, a lack of recognition was felt less from services—as there was no attempt to get support—but from the wider family network and sometimes the parent and siblings themselves. This could occur where older siblings have a greater understanding of the unmet needs of their siblings and the gaps in parenting due to having experienced similar parenting. While services were not involved, so the potential for care entry was not explicit, there was an acknowledgement that, if people knew what was going on, the children would be removed.

### 3.2.3. Group 3: Filling a Gap in Parenting

The last group is the one that most challenges the current understanding of what kinship care entails. This group was formed mainly from two narrative accounts—Hasan and Kelly. However, many of the narrative accounts in the previous group have similarities (the early parts of Laura's story show that she could be in this group). In these families, parents were still around in some capacity, technically in the same home, but not providing all the care that was needed or that the older sibling thought that their sibling needed. There is no acknowledged risk of care entry or sibling separation in this group, although there were accounts of abuse within the home. The older sibling (not always the oldest) then started to provide care for their younger sibling(s) (represented by route 5 on Figure 1).

These accounts highlight how siblings can carry out a caring role for their younger siblings. They might be seen as the main carers by their younger siblings, but they would not necessarily be viewed as such, not only externally by policies and services but also internally by family members.

Hasan's account shows how they came to feel that they were providing most of the care for their sibling. This was not recognised or validated by others.

> Hasan: *"My sister was 5 and I remember my mum and my sister and a couple of brothers that were living with her at the time, they all started to move because mum got evicted, there was no possibility of her finding anywhere else in that city. It was when she moved that I moved back in with my mum."*

*I think that's when things started in terms of sibling care. Because my mum has always been unreliable in every regard of her life. Especially when trying to bring up a child. She couldn't get the kids to school and that. My sister had really low attendance. She barely attended pre-school, nursery and she only just started attending reception but never went. So when they moved, that is when I took on full responsibility for my sister.*

*My mum was incapable of keeping a job, and I was working whilst doing my GCSEs to pay the rent, all the bills, to keep the house together. My mum would do all the stereotypically the women's job, housekeeping, keeping on top of the house…and I would do everything else. It was me, my mum, my sister and my little brother. Although I would care for my little brother, it was a lot less than what my sister needed. He was quite independent.*

*When it comes to my sister, when she moved, any difficulties with her school—I would be the one to deal with it. Any trips to be paid for, I would pay for because we weren't eligible for free school meals because of my dad, my dad made enough money. So ever since my sister moved, it was very much I did everything. When I was a young carer, I was 16, doing everything for my sister. It seems so impossible how I got through my A levels. It shouldn't have been possible. I was literally doing it part time, but it was a full-time course. I completely stopped talking to friends, I went to school, I sat there, did the coursework, did not speak to a single soul. I did triple the amount that a normal student would do, and then go home, and then I would have the rest of the day to sort of the house, then I'd walk an hour to go and get my sister and then pick her home by taxi, but I wouldn't pay for my taxi to go and get her because it was way too much money. At that time, I couldn't drive. It just seems impossible. It really isn't possible. But I managed to get through it.*

*I was reading about what a young carer is, I was like, 'Yeah, that's me. I'm doing this.' Then when I reached out for support, it was very much like, 'Oh, does your sister have a disability?' I'd say no, and then hang up. I think the one time I was offered support was when I when I explained my mother, and they were like, 'Okay, she sounds like she has a lot of mental health conditions. Are you a carer for her?' Which is the complete opposite. Although I personally think she has quite a lot of mental health conditions, she's never been diagnosed with it. There's a lot of stigma in our culture. She would never, ever, ever, go and get help for that. That was the only time where I was like, 'oh, you know, maybe here's some support' but it wasn't because I was caring for my sister, it was because my mother wasn't doing anything and was I caring for my mother.*

*I think that's where a lot of the identity crisis for me comes in".*

This is different from the lack of recognition experienced by other kinship carers. These carers do not fall under the current definition of kinship carers. They may not be recognised as young carers as they are not necessarily caring for a sibling with a disability or for a parent, although they would fit most definitions of a young carer. Where they are caring for a parent and siblings with and without disabilities, the multiple caring responsibilities may not be recognised. This can lead to these carers questioning their own identities and histories. This was echoed by Kelly.

Kelly: *"He [father] was in the house. But I think whether it was looking after or just physically, there. It's why me and him struggle with our relationship now, because he's not the best. But he was just physically there. In regards to actually looking after the siblings and making sure they're okay, that's kind of always been my job…. I find it difficult to categorise it because I know, I spent my whole childhood pretty much caring for other people".*

*But sometimes I'm sat down thinking maybe I've just made it up. Because…I don't know. Because my mum was there. Or because I did have a stepdad or. But then I'm like, they weren't doing anything".*

The narrative accounts in this group highlight children managing very serious and complicated safeguarding issues to keep themselves and their siblings safe, including physical and emotional abuse of the children, neglect of the children's physical and emotional needs, and domestic violence between adults in the home. Kelly described the role that she played for her parents and siblings, including providing for their basic needs, safeguarding them from domestic violence and witnessing inappropriate material, and being a "therapist" for her mother. She also describes having to build her own parents' parenting ability and "parent them in how to parent".

There are elements in these accounts that are very similar to the early stages of those in the other two categories, where older siblings provided care for their younger siblings throughout their childhood before 'becoming' kinship carer. These narrative accounts also look towards a future in which their roles might be similar to the other categories, with Hasan contemplating the formal adoption of his sister and Kelly imagining a future where she has to return home and take over the full-time care of her siblings.

> Kelly: *"Because I'm constantly drawn back. I'm constantly like, panicking. I'm constantly drawn back"*.

This complexity of shifting parenting responsibilities between kinship carer and parent is not unique to sibling carers. However, managing the parenting role in place of one's own parents while also living with and experiencing the same gaps in parenting is likely unique for this group of carers. These young people might reasonably be considered to be kinship carers if they are carrying out the main caring role for their siblings, and they certainly should be considered to be young carers, but there are potential barriers to their inclusion in either category due to inflexible definitions or a lack of understanding of the roles they are playing.

## 4. Discussion

The paper aimed to categorise and illustrate the experiences and journeys of sibling kinship carers when 'becoming' the main carer for their sibling(s). Narrative inquiry encouraged a focus on the holistic accounts of siblings to centre their experiences. This paper has presented three categories of accounts of 'becoming' a kinship carer by older siblings—remaking the family; keeping the siblings in the family; stepping in to fill a gap in parenting—based around the reasons they entered the role. Centralising the motivation of the carer helped to situate the agency of the participant in their experience rather than more traditional categories of kinship care, which focus on legal or statutory definitions or entitlements.

These narratives are helpful for understanding where families sit in relation to current understandings of policy and practice. However, they can only go so far in capturing the experiences of siblings. The small number of interviewees and the underrepresentation of male and non-white participants made it impossible to use a split-sample design to explore areas of interest such as gender or ethnicity. In addition, the exploratory nature of this study did not allow for a more in-depth focus on specific areas such as the impact of financial hardship or social support. However, this study provides an important starting point for future studies to explore these areas and compare this group's experiences with those of other carers.

Underpinning the accounts are the strong feelings and interdependent identities that siblings have in relation to each other. The development of identity in terms of the self (siblings' own sense of who they are) and identity (how they are viewed by others) [34] is a core part of the stories of sibling kinship carers. Some research points to how siblings' structure and restructure their stories based on how they are similar and different to their siblings and the importance of intra-generational relationships in forming identity [35–37]. However, the role of sibling carers in this research complicates the binary nature of intra- and inter-generational relationships, with sibling carers playing the roles of both carer and sibling. Often, research has focused on individual children and parent/child or grandparent/child caring relationships without acknowledging the significant role that

siblings have in each other's lives. Siblings are an important part of the family network and can be a source of safety and stability for children [38]. The role that siblings play in providing care and stability for each other in difficult home environments is not always sufficiently acknowledged or included when considering how a child can be cared for within their family network and the role that siblings play in child protection.

The narrative accounts reflect the long-term and multiple caring roles that older siblings often provide, showing how strongly these roles impact motivations and expectations to become kinship carer. All of the siblings 'stepped up' to provide care when it was needed. The accounts highlight numerous ways in which family members and practitioners may fail to see and appreciate the role that siblings play in developing a sustainable and safe family life for children. In Laura's account, the lack of financial support highlighted that the role she was playing was not recognised in any formal way. For others, lack of recognition from family and practitioners led to them questioning the importance of the role they were playing, especially where the parents were still somewhat present. In particular, the third group of carers illustrates how it is possible that a young person with a major role in the care of young children without disabilities may not be viewed as a kinship carer or within other definitions of 'carer', and therefore may not be deemed eligible for support. Understanding the impact of recognition and misrecognition can be an important starting point for practitioners involved in assessing and supporting sibling kinship carers [39]. Where the role of sibling carers has not been sufficiently recognised, this could impact their willingness to seek support when it is needed.

Some of the narrative accounts—particularly in the first two categories—highlight experiences and fears of separation through the care system. One large English study provided an estimate that 37% of children with a sibling who enter care are separated from at least one sibling at some point [40], although this figure fluctuates across the UK [41]. Research in kinship care indicates that between 54% and 88% of children in kinship care experience separation from one or more siblings [5]. Decisions to separate siblings may rely on preconceived categories such as gender, shared parents, and age rather than in-depth explorations of the nature and importance of the relationships for the children themselves [42]. Motivations to become kinship carers in response to separation or from fears of separation are therefore well founded and reflect other research highlighting a fear of children entering the care system as a key motivator to becoming a main carer [43].

This paper urges practitioners to consider the role of sibling relationships when deciding how to support children to remain within their family network, recognising the important role that siblings play in child protection. A lack of due care and prioritisation around maintaining sibling relationships can be a driver for sibling kinship care, with siblings who may be providing care excluded from decision making or from support through a lack of recognition. Moreover, the experiences that some kinship carers had in their own childhoods of social work involvement and the care system's needs could impact their decision making. For example, it is possible that some older siblings who have had negative experiences in care may make a decision to become carers for a sibling to protect them from entering care or from fear of separation.

Policymakers should consider how current policies and practices might disproportionately negatively impact sibling kinship carers, particularly in considering siblings as important partners in child protection. Moreover, attempts to establish an agreed definition of kinship care must grapple with complexities around who is carrying out the main caring role and differing perspectives on this, where there are shared caring responsibilities, and how all carers can be supported. The ambiguity surrounding the responsibilities of older siblings, who may assist at home or take on the primary caregiving role for their younger siblings, underscores the dynamic nature of caregiving roles within families. This complexity challenges simplistic notions of kinship care, revealing a more intricate picture of family life. Inclusive definitions of 'carer' and 'kinship carer' could explicitly recognise that some siblings might have major caring responsibilities even where there is another adult present. Finally, practitioners and those commissioning services should ensure services address the

specific needs of all of these groups of sibling kinship families and not allow definitions to limit access to support where it is needed.

## 5. Conclusions

This paper is not unique in arguing for more attention to be paid to sibling relationships for children who cannot be cared for by their parents. However, few studies have centralised the sibling relationship in kinship care, showing older siblings as active agents, motivated to provide care and support for their younger siblings. The paper also challenges current understandings, definitions, and practices that can exclude sibling kinship carers, disguising the role that they are and have played in the care of their younger siblings. Moreover, the accounts illustrate the importance of recognition for these carers, in helping them to become carers in the first place where this is an appropriate option and in enabling them to access the right support when it is needed. This may be particularly important where siblings are providing care but are not the only adults in the home, meaning the care work of these carers may be overlooked on the premise that the other adults are the primary carers of the children. It is hoped that these accounts will prompt practitioners and policymakers to "finally see" siblings when considering who is and should be involved in deciding how to support children to remain within their family network.

**Funding:** The research was funded by the ESRC under the ESRC Wales Doctoral Training Partnership (DTP) grant number ES/P00069X/1.

**Institutional Review Board Statement:** The study was conducted in accordance with the Declaration of Helsinki, and approved by the Cardiff University School of Social Sciences Research Ethics Committee (SREC/4093, approved 05.07.2021).

**Informed Consent Statement:** Informed consent was obtained from all subjects involved in the study.

**Data Availability Statement:** The datasets presented in this article are not readily available because the data is potentially identifiable even after pseudonymization. The data is stored securely at Cardiff University.

**Conflicts of Interest:** The author declares no conflicts of interest.

## Notes

[1] The 'organising theme' is at the heart of a narrative account, and can otherwise be thought of as the 'plot'. It is different from a theme within a narrative.

[2] All names are pseudonmyns.

[3] The PA acts as a focal point for the young person, ensuring that they are provided with the practical and emotional support they need to make a successful transition to adulthood, either directly or through helping the young person to build a positive social network around them. See: DFE (2018) Extending Personal Adviser support to all care leavers to age 25 Statutory guidance for local authorities (Avaliable online: https://assets.publishing.service.gov.uk/media/5a93ebb940f0b67aa5087986/Extending_Personal_Adviser_support_to_all_care_leavers_to_age_25.pdf accessed on 1 September 2023).

[4] In the UK, family members can be assessed to become a 'family and friends' or 'connected person' foster carer when a child is classed as 'looked after' under a care order. This involves an assessment against foster care compentencies, carried out by or on behalf of a local authority. See: What is kinship care? | CoramBAAF for different types of kinship care arrangements in the UK.

[5] In the UK, children who have been looked after under a care order are entitled to support through a 'leaving care team'. This support is in place for care leavers aged 21–25 depending on where in the UK they are, and when they were 'looked after'. See The-British-Academy-Young-People-Leaving-Care-A-Four-Nations-Perspective+(1).pdf (accessed on 1 September 2023) for differences between devolved nations.

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
