# Peer review of "“They Finally See Me, They Trust Me, My Brother’s Coming Home” Recognising the Motivations and Role of Siblings Who Become Kinship Carers"

_societies, doi:10.3390/soc14020024_

Round 1
Reviewer 1 Report
Comments and Suggestions for Authors
Summary
This is an interesting piece of qualitative research that builds on British work undertaken over 10 years ago, extending knowledge in an under-researched area of kinship care. The previous British qualitative research on sibling kinship carers was published as grey literature only, that is, in research reports rather than refereed journal articles. While the current study is small, if it is published, it may be the first British qualitative research on this topic to appear in a refereed journal. The manuscript needs revision to be publishable.
Vague or imprecise language
An issue with the overall presentation of the manuscript is that the use of language is sometimes clear and precise, and at other times, unclear. I have identified only a few examples; the whole manuscript needs to be reviewed to improve clarity and precision of language. While the writer’s command of English is good, they might consider working with a ‘critical friend’ or alternatively, using an editor to improve precision and clarity.
Page 2, line 59. No reason is given to justify the suggestion presented in the first sentence in this paragraph.
Page 4, line 158. The term ‘organising theme’ used in quotation marks would be clearer without the quotation marks, or could be simply replaced by theme, as in:
Narratives were grouped according to three identified themes relating to these carers’ motivations to provide care to their siblings.
If the writer doesn’t see it as dehumanizing, the groups could be referred to as Group 1, Group 2 and Group 3 in the reporting of results.
I suggest replacing the two uses of the word ‘pathways’ with ‘motivations’ or ‘motivational paths’ to increase clarity about the themes.
Page 4, line 163. The concept of a ‘supporting cast’ is mentioned here, but it is unclear why the concept has been introduced as there seems to be no follow through about it.
Page 4, line 167. The notion of ‘ ”problematising’ the common categorization of ‘informal’ and ‘formal’ kinship care arrangements” is not spelt out. The writer should explain what they identify as the problem with these terms.
Typos and inaccurate choice of words
There are a few words missing here and there, and a few small inaccuracies in words selected. Such issues are readily rectified; each line needs to be reviewed carefully to identify these. Just two examples are:
Page 1, line 34. I think the accurate word here would be ‘carers’ or ‘kinship carers’, not ‘families’.
Page 5, line 185.
These siblings were informed that their siblings had entered care and were involved in thinking about the next steps.
It is not clear which individuals were referred to as being involved in thinking about the next steps. It seems unlikely that the younger siblings were being invited to do so. Is the writer referring by implication to the welfare authorities involved in the care arrangements?
Over-generalisation
There are instances of over-generalisation throughout the manuscript. I looked up a few of the references to check this, hence some examples provided below. The writer needs to check all of their references to ensure that citations are accurate.
Research findings should be cited in the past tense to limit findings to the particular time, place and methodology of each study; other studies under different circumstances might produce different results. The author needs to be particularly careful generalizing from small, non‑random samples. There are a number of such over-generalisations in the literature review, just 2 examples being:
Page 2, line 49. ‘Kinship carers report feeling…’ which might be better expressed as ‘A few studies identified reported that some kinship carers felt…’
Page 2, line 68. ‘Potentially around a third of the children in informal kinship care may have experienced parental bereavement…’ This is followed by a speculative comment (line 69 onwards) about bereavement possibly being more common for children living with siblings than with grandparents, a comment that appears to lack an evidence base. While the first statement has been made tentatively, it nevertheless overstates the data from non‑random, self‑selected populations that it draws on. This first statement would be better expressed along the lines of:
In two unpublished studies of kinship care [8,17], around one-third of children had experienced parental bereavement.
The second speculative statement would be better removed.
Page 2 line 42. 6.25% of kinship carers are noted as being siblings in the study cited [reference 8]. This figure is accurate. However it is potentially misleading as this was actually a very small group of 5 sibling carers, being 6.25% of the total study sample of 80. The writer might mention the total sample size for to make this clear.
Examples (among others) in the Discussion:
Page 13, line 388. ‘Some research points to’ appears to be a reference specifically to citation [32], noted as ‘see for example [32]’. Unless 32 is a review article, this statement would be better supported by more than one reference, in which case it could be cited directly rather than as ‘see for example’.
Page 14, line 413. The author asserts there are widely varying estimates of sibling separation in care, stating that 37-74% of British children with a sibling who enter care are separated from at least one sibling at some point. The figure of 37% derives from an analysis of 20,000 British children in care, thus a large study that provides a good basis for an estimate. I searched in the other article cited for the figure of 74%, but as far as I could determine, this study actually reported a figure of 86%. Apart from a possible misquote of the figure, this study does not provide a good basis for generalisation as the total sample was only 49 children. A simpler statement would be more appropriate, e.g.:
One large British study provided an estimate that 37% of children with a sibling who enter care are separated from at least one sibling at some point.
The writer should review all citations of facts and figures to ensure accuracy and appropriateness of assertions based on the research literature.
Page 14, line 401. To avoid over-generalisation, substitute ‘may’ for ‘often’.
Page 14, line 403. To avoid over-generalisation, add the word ‘may’ between ‘practitioners’ and ‘fail’.
Comments on specific sections of the manuscript
Abstract
The use of the term ‘categories’ in quotation marks suggests to me that a better term could be identified. The ‘categories’ referred to here are very small groups of carers whose motivations have varied. Consider using the term groups or groupings, perhaps referring to the groups or groupings of narratives that identified three different motivational paths to becoming a kinship carer.
Introduction
The literature review has identified some key publications relevant to the topic of research. The author is to be commended for having found the relevant grey literature. One other relevant publication that I am aware of is:
Denby, R. W. and J. Ayala (2013). "Am I My Brother's Keeper: Adult Siblings Raising Younger Siblings." Journal of Human Behaviour in the Social Environment 23(2): 193-210.
Materials and methods
I think the first two paragraphs of this section (Page 2, lines 86-103) would be better placed as the last two paragraphs of the previous section.
Page 2 line 87. It is suggested here that the perspective of the younger siblings was part of this study, but it was apparently not as it is not reported here. It would be preferable to delete this reference. If the author is referring to a different aspect of their PhD study that is not the subject of this manuscript and wants to allude to it, this should be made clear.
Page 3 line 109
The author states: ‘Adopting an approach that does not necessitate specific sample sizes or strict inclusion criteria allowed for the exploration of as many pathways to kinship are to be included as possible.’
First, I do not understand the reference to the requirement of other research processes to ‘necessitate specific sample sizes’, as qualitative research methodologies do not do that. I also think it is difficult to argue that a more flexible approach ‘allowed for the exploration of as many pathways to kinship are to be included as possible’, given that the total sample size was 13, and the identification of 3 pathways (or motivational paths) to kinship care does not suggest exploration of ‘as many pathways to kinship care’ as possible.
Page 3, line 119. Explain what ‘the pre-task’ was, or delete the reference to it.
Page 3, lines 132 onwards. The description of the thematic analysis utilized seems very general; I am more clear about how the analysis was not conducted than about how the themes were actually identified. Codes used in qualitative analysis do not have to be pre-determined; in a grounded theory approach they are generated from the exploration of the data itself, ‘ground up’. Was any coding used in this study? If not, what process was actually used to identify the three motivational paths as the themes of the findings?
Results
3.1 Participants
I would like to see some brief demographic data presented at the beginning of this section. Reiterate the total sample size here. How many were older sisters, how many older brothers? If the data was collected, note how many carers were single and how many partnered; the age range of the carers with any particular features (e.g. a cluster of particularly young carers); how many children each was caring for; and any other notable features within the cohort such as ethnicity. I suggest noting specifically that the majority (8) out of the total of 13 subjects had no legal order in place, rather than referring to them as ‘the remainder’. Such data could be presented in a Table, with notable features highlighted in the text.
The connection between legal orders or lack of them and the three motivational paths identified is illustrated in Figure 1. While I recognise the writer does not wish to emphasise the formal/informal categorisation, it is relevant to mention it in the text in relation to the themes of study, and the specific numbers might be superimposed onto the circles representing each thematic grouping in Figure 1. My impression is that the first thematic grouping included 5 carers’ families, the second grouping 6, and the third grouping 2; this could be spelt out.
3.2.1 Remaking the family
Unlike the other carers presented in this section, there seemed to be almost no analysis of Joanne’s story, the story most fulsomely presented verbatim. I am not sure why.
More specificity is needed in some of the findings, e.g.:
Page 7, line, 246-247, is the writer commenting on 2 carers’ stories? This should be spelt out, possibly removing the word ‘often’ in this sentence.
In line 255, more specificity is needed in relation to ‘Other narratives highlight…’ How many narratives? If a small number, the relevant carers could be mentioned by name.
Page 5, line 198. Who is meant by the older sibling’s personal advisor? Is this a friend, or someone designated by the care authority?
Page 6. Joanne’s story. The verbatim narrative needs either a couple of more precise explanatory comments in the square brackets, or a brief introduction outlining who the family members are. Is ‘[their dad]’ the same person as ‘[the younger children’s father]’? Who is Andrew? Is ‘[their nanny]’ the same person as ‘nan’?
Page 7, line 237. It would be helpful if a few words in square brackets could explain what photo Sally was showing the interviewer and why.
For ease of reading, the introduction to Sally’s story might be better started on a new paragraph; this could still refer back to a commonality with Joanne’s story with regard to shared family histories.
3.2.2 Keeping the siblings in the family
Page 9, Laura’s story. This narrative is again a little confusing without any explanatory comments. Is ‘my big brother’ the same person as ‘My older brother’?
3.2.3 Filling a gap in parenting
This section highlights a grey area in current terminology by featuring two carers in households where parents are residing but providing little parenting. A bit of revision of the text here might improve articulation of the ambiguity in current definitions and the problems this generates. Rather than asserting that these two siblings were actually kinship carers, the writer could suggest that they might reasonably be considered to be kinship carers, and that they certainly should be considered to be young carers, but that there are potential barriers to their inclusion in either category. It would appear that perhaps the British definition of a ‘carer’ does not include kinship carers, as is the case where this reviewer lives and works. More could be made of the point that a young person with a major role in the care of young children without disabilities may be excluded from young carer services. Also, that the definition of a kinship carer may need clarification. Does it assume there are no other adults living in the home? If others are living in, how much care of the household’s children by people other than the parents is sufficient to be identified as the kinship carer? Perhaps with more inclusive definitions of ‘carer’ and ‘kinship carer’, the potential of a blurred boundary between understandings of ‘young carers’ and ‘kinship carers’ might not matter. The author is thus raising an important point about the difficulty at this interface and the adverse effects on young people with major caring responsibilities of falling between the cracks in terms of eligibility for service and support. (Clearly, the writer has here touched a point of some concern to this reviewer!)
Page 13, line 354. Domestic violence is mentioned here for the first time. I suggest bringing this topic in earlier in the article where relevant, or deleting it.
Page 13, line 363. I was surprised to detect an inference that adoption of a sibling might be legally possible in the UK. Is this really the case?
Discussion
Page 13 line 374. The reference here to a purposive sampling approach to explore a range of different pathways to care came as a surprise. If a purposive sampling approach was utilized, this should be described in Section 2 Materials and Methods. How was the purposive sampling approach carried out? Did the researcher have possible different motivational pathways in mind? If so, did they approach particular organizations or services assuming that they might be involved with carers who had particular motivational paths to care? For example, I am guessing that perhaps statutory child protection authorities might give access to carers who were motivated to ‘Remake the family’ by bringing children out of foster or residential care.
Page 14, line 423. Does ‘decisions being made in fear rather than partnership’ refer to older siblings’ decisions, or to social workers’ decisions? Is the reference to social workers’ fear that young people can’t be competent parent figures, or to young people’s fear of what might happen to their siblings if they don’t take on their care?
Page 14, lines 424 onwards. The last sentence needs elaboration. In what way might the experience of social work involvement in the prospective carer’s childhood impact on older siblings’ decision‑making?
Conclusions
Page 14, line 429. Delete ‘birth’ from ‘birth parents’. Parents are parents; the word ‘birth’ is discriminatory and unnecessary.
Page 14, line 435. Replace ‘where this is the right option’ with ‘where this is an appropriate option’.
Presentation
The physical layout of the manuscript generated some difficulty with appreciating its overall shape, compromising ease of reading.
Presentation of verbatim narratives
Verbatim narratives are presented inconsistently.
Ø The three long verbatim narratives are presented in boxes. In each case, some of the text appears to have disappeared below the frame of the box and therefore could not be read. I do not know how much text was unavailable for review. In the resubmission I suggest a different mode of presentation – not boxes.
Ø On page 7, some verbatim narrative from interviewees is presented in separate paragraphs in Italics which makes for clear reading, but shorter pieces are included as run-ons within the text which is not easy to read.
Ø On page 10, verbatim narrative is presented in separate paragraphs, but not in Italics. Double quotation marks within verbatim narratives are used inconsistently.
Reading would be easier if all verbatim material from interviewees was presented in one consistent clear format, such as by indenting all verbatim narrative. If double quotation marks are used and further quotation marks are needed within the quote, single quotation marks should be used.
The figure
Figure 1 is too small to be read easily and I was thus unable to evaluate it fully. It would be better presented as a full-page figure. Unless the journal can print Figures in colour, it might be appropriate to see if it could be presented clearly in black and while.
Citations
The system of citations is not consistent. The author has mostly used a numeric style of referencing, but later in the article there are a couple of author, date citations.
Comments on the Quality of English Language
The writer's command of English is technically excellent, however the text lacks clarity and precision in places. See detailed review for more discussion of this issue.
Reviewer 2 Report
Comments and Suggestions for Authors
Dear Authors:
It is an exploratory and very interesting manuscript. I miss the ethical considerations of the study.
Regards
Author Response
Thank you for the kind review. I have ensured that the paragraph on ethical considerations is clearer.
Reviewer 3 Report
Comments and Suggestions for Authors
The strength of this paper is that study findings clearly illustrate the complexities in family life that lead to sibling kinship care, and opens up understanding of the specific rationale, purpose, and goals that older siblings may employ in taking responsibility for their younger siblings care. The paths to sibling kinship care revealed through the research will be informative for services- recruitment and and support of sibling kinship care. Services designed and delivered can be more in tune with the particular circumstances associated with the sibling kinship care set up.
Under materials and methods, the authors should provide a little more details about what is meant by "think about their life in five chapters or events" ( lines 123-125).
The authors should provide more details about the interviews- What was the " ' experience near' " question used; were there any other follow-up questions to guide the narrative interview, given the emphasis on "reducing the cognitive load on participants".
How long did each interview last? Were the interviews conducted by the same person? Was the analysis conducted by one person? Were there any strategies of rigor employed during data collection and analysis?
The author should highlight more directly the implications of the findings for training and education of service providers engaged in kinship care programming.
Reviewer 4 Report
Comments and Suggestions for Authors
I generally like this study of kinship care in the UK. I have some friendly recommendations for improvement through revision.
1. Please provide the strongest possible justification for this study. It seems that kinship care arrangements are declining, which is an important fact but could undermine the study rationale (fewer people affected). No need to retract that, but add something perhaps. The understudied nature of the topic is good, but the case could be made that we can learn something unique and valuable about family life and contemporary family relationships in particular from studying unusual cases like this. Consult and cite research supporting extreme case analysis. This seems to fit that description.
2. How was the choice made to interview 13 individuals? In many cases, 20 interviews are common. This project could be described as a pilot study, which does not undermine its significance but does set appropriate expectations concerning interviewee numbers. Also, the family type studied here is relatively rare, so that might influence interviewee availability.
3. More detail on the cross-national approach to interviewing might also be provided. Do the laws about kinship care vary across these nations?
4. After three decades of conducting qualitative research, I am convinced that theory is an ally of qualitative researchers. Consider adding a paragraph or two on theory and then revising (not overhauling) the results in light of the theory. My recommended candidate would be "doing family" theory. Family adaptive strategies might also be a useful theoretical lens. Either or both of these frameworks would complement and even sharpen the methodological approach used here and reporting of results by emphasizing how family practices and negotiated interactions that emerge from interview narratives can be most soundly analyzed. I am not a fan of a thematic (grounded theory) approach to transcript analysis. Theory can enrich the results and explanations rendered through them.
5. More detail on the analytical procedures would also be welcome. Thematic analysis following the use of guiding concepts derived from theory is certainly acceptable. More to the point, were the transcripts read multiple times and how many researchers conducted them? Was coding conducted manually or with the use of qualitative data analysis software? Walk us through the analytical phases.
6. Study limitations should highlight the small number of interviewees and subsequent inability to use a split-sample design (e.g., comparative analyses by gender given the femininization of care work). That could be a fruitful direction for future research. Be more clear about study limitations and how they might be addressed by others going forward.
Small items: Avoid single-sentence paragraphs. In technical writing, a paragraph typically is 4-6 sentences. Also, some item such as funding, IRB, and acknowledgments need to be addressed.
Comments on the Quality of English LanguageGenerally sound, and could benefit from a careful proofreading.
Round 2
Reviewer 1 Report
Comments and Suggestions for Authors
As mentioned in my first review, this paper is of interest as a report of a piece of research in an under-researched area. The layout of the paper is now clearer. There has been significant rewriting and useful additional material about methodology, and the sample demographics. It would still be helpful to be specific about kinship carer gender in the text (presumably 12 young women and 1 young man), and in the Table, the number of children each kinship carer was caring for at the time of interview.
However, issues that compromise the coherence and clarify of the paper remain. I would like to see this material published, but I now consider that in preparing another draft the writer needs both further detailed advice from his thesis supervisor, and the services of a professional editor. Below are just a few of many examples of issues that need to be addressed.
1. Referencing
There is a major problem with the referencing. Looking at some of the citations, I wonder whether the numbering system has gone out of kilter, as some citations refer to articles that could not possibly contain the material cited. In some instances, the correct reference may be the one following the cited reference in numerical order. Others appear to be simply inaccurate. Three examples from one short section on page 2 follow plus one from page 4.
· Page 2 line 42 (first line on the page) states: 'Stories powerfully communicate real world experiences in the context of social work [10] and influence policy. I cannot see anything in the cited article that justifies this citation.
· Page 2 line 48 includes reference [6] as a citation of a qualitative study that 'focused solely on sibling carers (n=12)'; however reference [6] is a systematic review. I think this citation should be to reference [7].
· Page 2 line 51. The writer states: 'Research from elsewhere in the world (i.e. [7, 8; 13]) resonate with findings from the UK...' However, references 7, 8 and 13 are all UK papers. Note also that ‘e.g.’ should be used instead of ‘i.e.’.
· Page 4, line 164. There is a citation [27: p58], referencing the methodology employed. However citation [27] refers to an article with page range pp.328-352. Reference [26] directly above in the Reference List has page range pp.85-94, and reference [28] directly below has page range pp.226‑236. None of these three references (which all address methodology) seem to have a page 58.
2. Language
More critically, the written language is still often vague or imprecise. One aspect is the use of words in quotes, which needs to be reduced as far as possible. Replacing the word in quotes with one or more carefully chosen words can provide greater clarity of meaning. A few examples of many other issues with clarity follow.
· Page 1 line 31. What is the point being made here about the collection of statistics? The statistical definitions may need to be explained. It would help to provide a small amount of detail about the way population level statistics are collected as outlined in references [3] and [4] cited above on line 30.
· Page 3. Line 98 onwards. This paragraph is a positive addition to the paper, identifying the lived experience of the writer. As mentioned in the first review, it should also specify that this particular paper addresses one component of the PhD project, being that of the voice of the sibling carers themselves. Otherwise the reader is left looking for the voices of the younger siblings, and of the ‘key stakeholders and practitioners’.
· Page 4 line 170. What is meant by ‘timelines’ and ‘restory’ in quotes? If the word timeline is precise, it does not need to be in quotes. I think the term ‘restory’ needs explanation, as otherwise it might suggest a questionable reinterpretation of the subject’s narrative.
· Page 5. The writer has clarified and justified the use of the term organising theme rather than simply theme. He should now review his use of the term theme throughout, as it may be that at some points theme with organising theme may be better. Thematic category in the Table, for example, might better be expressed simply as Organising theme.
· Page 8. Rather than the new footnote about a picture in response to my earlier comment, it would be more helpful to have a couple of words in brackets to explain what the picture was, e.g. [photo of carer with children].
· Page 10. Top line of page.
“The narratives in this group problematise the idea of ‘formal’ kinship families as being ones with social work involvement, and ‘informal’ kinship families having made a kinship arrangement themselves.”
The problematisation being referred to is not clear. Is it the point that many ‘informal’ kinship families have had local authority social work involvement but have not become ‘formal’ kinship arrangements? This is a reasonable point, but needs to be clarified as the problem, or the term problematisation replaced.
· Page 14, first line of page plus footnote. This footnote has also been added in response to a query in my first review about whether adoption is possible for kinship carers in the UK. However the footnote relates to Scotland, and the kinship carer being referred to here was in England, so the writer has had to add yet another additional qualifying comment in the footnote. On reflection I think commenting on this matter is more confusing than clarifying, so I suggest removing the footnote. (Sorry for this!)
· Page 16, Conclusions:
“This may be particularly important where siblings are providing care but are not the only adults in the home.”
Spell out the point being made here, which is presumably that the care work of these kinship carers/carers may be overlooked on the premise that the other adults are the primary carers of the children.
· Page 16, Conclusions:
…’finally see’ is written in quotes. Use a clearer word without quotes. Perhaps notice?
3. The issue of definitions limiting recognition and support
Page 10 line 388-391. By the use of ‘becoming’ in quotes, the writer has missed an opportunity to be clear about the important point that he is presumably making. Instead of:
“In the illustrative narrative account below, Laura talks about how she had been caring for her brother in the years before ‘becoming’ a kinship carer.”
I would suggest something like:
In the illustrative narrative account below, Laura talks about how she had been caring for her brother in the years before a crisis led to her role being recognised as that of a kinship carer, her role however not subsequently being formalised, leading to a lack of support.
This example highlights what I think is a key feature of this paper as mentioned in my first review – that is, identification of ambiguity in the terms kinship carer and carer and the associated problems for people in caring roles of various natures. Despite some rewriting, I think this issue could still be highlighted and discussed more clearly so that it might contribute to improvement in definitional accuracy in policy and practice for both kinship carers and carers, especially in this case, young people in caring roles. In my mind at least, this might include a desirable and logical move to include kinship carers in the broader definition of carers. (This matter is the subject of advocacy in my country at the present time.) The writer has picked this issue up at various points in the paper (middle of page 11; middle of page 13; and in the Discussion). The Discussion is the place to pull this material together and discuss it specifically. Refer also back to my earlier comments on this issue in the first review. While challenging, I think work on this aspect of the paper may yield dividends for its overall significance.
Comments on the Quality of English LanguageSee above.
Author Response
Thank you for your comments, I have responded to each in the attached word document.

Reviewer 4 Report
Comments and Suggestions for Authors
I commend the authors on a sound and comprehensive revision.
With women so overrepresented, how about integrating some more work on the gendered nature (feminization) of care work? Some attention to this in the results and discussion seems warranted.
Comments on the Quality of English LanguageStandard proofreading recommended.
Author Response
With women so overrepresented, how about integrating some more work on the gendered nature (feminization) of care work? Some attention to this in the results and discussion seems warranted.
Thank you for your comments. This is certainly an area that I would like to explore further and is important in this group of participants. However, it feels like to do this justice would be out of the scope of this paper. There is some interesting work that indicates different trends for different age ranges of children (i.e. Kiraly 2021) which I would be keen to explore further. I have not added this in due to the limits in the word count. Although if it is felt to be necessary, I would be more than happy to do this. I am aware of the small sample I have so do not want to over generalise from this (as per reviewer 1's comments).